# Modeling of One-Side Surface Modifications of Graphene

**DOI:** 10.3390/ma12244179

**Published:** 2019-12-12

**Authors:** Alexander V. Savin, Yuriy A. Kosevich

**Affiliations:** N.N. Semenov Federal Research Center for Chemical Physics of Russian Academy of Sciences, 4 Kosygin str., 119991 Moscow, Russia; yukosevich@gmail.com

**Keywords:** graphene, one-side modification, hydrogenation, fluorination, chlorination

## Abstract

We model, with the use of the force field method, the dependence of mechanical conformations of graphene sheets, located on flat substrates, on the density of unilateral (one-side) attachment of hydrogen, fluorine or chlorine atoms to them. It is shown that a chemically-modified graphene sheet can take four main forms on a flat substrate: the form of a flat sheet located parallel to the surface of the substrate, the form of convex sheet partially detached from the substrate with bent edges adjacent to the substrate, and the form of a single and double roll on the substrate. On the surface of crystalline graphite, the flat form of the sheet is lowest in energy for hydrogenation density p<0.21, fluorination density p<0.20, and chlorination density p<0.16. For higher attachment densities, the flat form of the graphene sheet becomes unstable. The surface of crystalline nickel has higher adsorption energy for graphene monolayer and the flat form of a chemically modified sheet on such a substrate is lowest in energy for hydrogenation density p<0.47, fluorination density p<0.30 and chlorination density p<0.21.

## 1. Introduction

Recently, intensive studies have been performed of various derivatives of graphene (hexagonal monolayer of carbon atoms) [1,2,3,4], such as graphane CH and fluorographene CF (a monolayer of graphene, completely saturated on both sides with hydrogen or fluorine) [5,6,7,8], grafone C_2_H (graphene monolayer saturated with hydrogen on one side) [9,10,11,12], one-side fluorinated graphene C_4_F [13,14], chlorinated graphene C_4_Cl [15]. Valence attachment of an external atom to a graphene sheet leads to the local convexity of the sheet as result of the appearance of the sp^3^ hybridization at the joining point [16,17]. Therefore, if hydrogen atoms are attached on one side in the finite domain of the sheet, creating a local peace of graphone on the sheet, a characteristic convex deformation of the sheet occurs in this region [18]. If the hydrogen, fluorine or chlorine atoms are attached uniformly to one whole side of the sheet, the whole small sheet will take a convex shape while a large sheet will fold into a roll [19,20,21,22].

From the macroscopic point of view, the bending of a graphene sheet under the effect of one-side chemical modification is similar to the bending of a thin solid film caused by the difference of surface stresses on its sides. Such difference is created during the thin film growth on a substrate with lattice mismatch [23] or during the one-side epitaxial growth of a surfactant on the film [24,25]. This effect is used, for example, for the measurements with optical technique of the change in surface stress caused by the monolayer and sub-monolayer adsorption of a surfactant [26,27]. On the other hand, the equal surface stresses on both sides of elastic thin film cause the change of the film thickness inversely proportional to its thickness [28,29,30,31], which can also be used for the characterization of the modification of film surfaces state induced by surface treatment. Corrugated conformation of a graphene sheet can also be controlled by the patterned vacancy ’defects’ in its hexagonal lattice [32].

The most convenient way to obtain graphene sheet modified on one side is to attach the sheet to a flat substrate and further chemically modify its outer surface. For getting a graphone sheet, the latter needs to be hydrogenated being attached to a substrate. Modeling of the hydrogenation of a graphene sheet [33,34] has shown that the substrate has a significant effect on this process, and it is very difficult to obtain a perfect graphone-like structure with the formula C_2_H (one hydrogen atom per two carbon atoms). Hydrogenation leads to the formation of randomly distributed uncorrelated domains with average hydrogenation density of the sheet p<0.5.

Important experimentally observable consequence of the coupling of two-dimensional atomic layer with elastic substrate is the appearance of the gapped resonance modes in the vibrational spectrum of the two-dimensional system of distributed oscillators on elastic substrate [35,36,37,38,39,40]. In Section 2 we use the value of the spectral gap in transverse oscillations of the monolayer on elastic substrate for the evaluation of the coupling strength of carbon, fluorine and chlorine atoms with nickel substrate.

In this paper, we model with the use of the force field method, the dependence of the mechanical conformations (formation of secondary structures) of graphene sheets, placed on flat substrates (flat surfaces of molecular crystals), on the density of one-side attachment of hydrogen, fluorine or chlorine atoms.

## 2. Model of Modified Graphene Sheet

In our modeling, we use the force field AMBER (Assisted Model Building with Energy Refinement) [41]. To model the chemically modified graphene sheet, we use the force field in which distinct potentials describe the deformation of valence bonds and of the valence, torsion and dihedral angles, and of non-valent atomic interactions [42]. In this model, the strain energy of the valence sp^2^ and sp^3^ C–C and C–CR bonds, and of O–H, C–R bonds (here an atom or group of atoms R = H, F) is described by the Morse potential:(1)$$V\left(\rho \right)=\epsilon_{b}{\left[e^{-\alpha (\rho -\rho_{0})}-1\right]}^{2},$$
where ρ and $\rho_{0}$ are the current and equilibrium bond lengths, $\epsilon_{b}$ is the binding energy, and the parameter α sets the bond stiffness $K=2\epsilon_{b}\alpha^{2}$. The values of potential parameters for various valence bonds are presented in Table 1.

Energy of the deformation of the valence angles X–Y–Z is described by the potential
(2)$$U\left(u_{1},u_{2},u_{3}\right)=U\left(\varphi \right)=\epsilon_{a}\left(\cos\varphi -\cos\varphi_{0}\right),$$
where the cosine of the valence angle is defined as $\cos\varphi =-\left(v_{1},v_{2}\right)/|v_{1}\left|\right|v_{2}|$, with vectors $v_{1}=u_{2}-u_{1}$, $v_{2}=u_{3}-u_{2}$, the vectors $u_{1}$, $u_{2}$, $u_{3}$ specify the coordinates of the atoms forming the valence angle φ, $\varphi_{0}$ is the value of equilibrium valence angle. Values of potential parameters used for various equilibrium valence angles are presented in Table 2.

Deformations of the torsion and dihedral angles, in the formation of which edges carbon atoms with attached external atoms do not participate (torsion angles around sp^2^ C–C bonds), are described by the potential:(3)$$W_{1}\left(u_{1},u_{2},u_{3},u_{4}\right)=\epsilon_{t,1}\left(1-z\cos\phi \right),$$
where $\cos\phi =\left(v_{1},v_{2}\right)/|v_{1}\left|\right|v_{2}|$, with vectors $v_{1}=\left(u_{2}-u_{1}\right)\times \left(u_{3}-u_{2}\right)$, $v_{2}=\left(u_{3}-u_{2}\right)\times \left(u_{3}-u_{4}\right)$, the factor z=1 for the dihedral angle (the equilibrium angle $\phi_{0}=0$) and z=−1 for the torsion angle (the equilibrium angle $\phi_{0}=\pi$), the energy $\epsilon_{t,1}=0.499$ eV (the vectors $u_{1}$,…,$u_{4}$ determine equilibrium positions of the atoms, which form the angle). More detailed description of the deformation of the torsion and dihedral angles is given in [43].

Deformations of the angles around sp^3^ bonds C–C^′^ are described by the potential:(4)$$W_{2}\left(u_{1},u_{2},u_{3},u_{4}\right)=\epsilon_{t,2}\left(1+\cos3\phi \right),$$
with energy $\epsilon_{t,2}=0.03$ eV.

It is worth mentioning that the attachment of two hydrogen atoms on one side of the sheet to the carbon atoms bonded by valence bond is not energetically favorable [44]. Therefore, we will consider such attachment configurations on one side of the sheet of X atoms (X = H, F, Cl), in which if an X atom is attached to one carbon atom, the X atoms are not attached to the three neighboring carbon atoms. The valence bonds CX–CX, and the valence and torsion angles formed by these bonds, are absent in such structures. Therefore the corresponding potentials can be omitted.

The nonvalent van der Waals interactions of atoms are described by the Lennard-Jones potential
(5)$$W_{0}\left(r\right)=4\epsilon_{0}\left[{\left(\sigma /r\right)}^{12}-{\left(\sigma /r\right)}^{6}\right],$$
where *r* is the distance between interacting atoms, $\epsilon_{0}$ is the interaction energy (equilibrium bond length $r_{0}=2^{1/6}\sigma$). The used values of the potential parameters for different pairs of atoms are presented in Table 3. The values of the potential parameters for carbon atoms of the graphene layer are taken from [45], for the remaining atoms are taken from [46].

In the simulation, the polarization of the C–F and C–Cl valence bonds was taken into account. At the atoms forming these bonds, the charges −q, +q were used from the PCFF force field (for the first bond, the charge q=0.25e, for the second bond q=0.184e). The interaction of two hydroxyl groups (hydrogen bond OH⋯OH) was described with the use of the potentials from the PCFF force field.

We define the interaction of the sheet with a flat substrate using the potential $W_{s}\left(h\right)$, which describes the dependence of the atomic energy on its distance to the substrate plane *h*. For a flat surface of a molecular crystal, the energy of the interaction of an atom with a surface can be described with a good accuracy by the (k,l) Lennard-Jones potential [47]:(6)$$W_{s}\left(h\right)=\epsilon_{s}\left[k{\left(h_{0}/h\right)}^{l}-l{\left(h_{0}/h\right)}^{k}\right]/\left(l-k\right),$$
where l>k is assumed for the exponents. The potential (Equation 6) has a minimum value $W_{s}\left(h_{0}\right)=-\epsilon_{s}$ ($\epsilon_{s}$ is the binding energy of an atom with the substrate). For a flat surface of crystalline graphite, the exponents in the potential are l=10, k=3.75. The binding energies are $\epsilon_{s}=0.052$, 0.0187, 0.0465 and 0.1026 eV for the C, H, F, and Cl atoms, respectively, the corresponding equilibrium distances are $h_{0}=3.37$, 2.92, 3.24, and 3.435 Å.

When graphene is located on the surface of crystalline nickel, a stronger chemical interaction of carbon atoms with the atoms of the substrate occurs. Therefore, the interaction of the carbon atom in the graphene sheet with the (111) surface of the Ni crystal is more convenient to describe by the Morse potential:(7)$$W_{s}\left(h\right)=\epsilon_{s}{\left\{\exp\left[-\beta \left(h-h_{0}\right)\right]-1\right\}}^{2}-\epsilon_{s}.$$

For a carbon atom, the interaction energy with the nickel surface is $\epsilon_{s}=0.133$ eV [48] and the equilibrium distance to the substrate plane is $h_{0}=2.135$ Å [49].

In result of the interaction of a graphene sheet with a crystal surface, a gap of the magnitude $\omega_{0}=240$ cm^−1^ appears at the bottom of the frequency spectrum of transverse oscillations of the sheet [40]. From this we can estimate the harmonic coupling parameter of the interaction of the sheet atom with the substrate $K_{0}=\omega_{0}^{2}M=41$ N/m (*M* is a mass of carbon atom), see also [35], as well as the value of the parameter $\beta =\sqrt{K_{0}/2\epsilon_{s}}=3.1$ Å^−1^. For the fluorine atom we obtain $\epsilon_{s}=0.13$ eV, $h_{0}=1.655$ Å, β=3.75 Å^−1^, for the chlorine atom we obtain $\epsilon_{s}=0.299$ eV, $h_{0}=2.115$ Å, β=3.17 Å^−1^.

In the following we will consider only these two substrate potentials. The first potential describes the weak interaction of the sheet with the substrate while the second potential describes the strong interaction. Other commonly-used substrates (surfaces of crystalline silicon Si, silicon dioxide SiO_2_. silicon carbide 6H-SiC, hexagonal boron nitride h-BN and gold Au) are characterized by the intermediate values of the coupling parameters—see Table 4.

## 3. Stationary Structures of Square Sheet

To find the stationary state of the modified graphene sheet, it is necessary to find the minimum of the potential energy
(8)$$E\to \min:{\left\{u_{n}\right\}}_{n=1}^{N},$$
where *N* is the total number of atoms on the sheet, $u_{n}$ is a three-dimensional vector defining position of the *n*th atom, *E* is a total potential energy of the molecular system (given by the sum of all interaction potentials of atoms in the system). The minimization problem (Equation 8) will be solved numerically by the conjugate gradient method. Choosing the starting point of the minimization procedure, one can obtain all the main stationary states of the modified sheet bonded with a flat substrate.

Consider a square graphene sheet of size 8.47×8.37 nm^2^, consisting of $N_{c}=2798$ carbon atoms. The sheet has $N_{b}=148$ edge atoms. To simplify the model, we assume that only one hydrogen atom is always attached to each edge carbon atom, see Figure 1a. With maximum one-side hydrogenation of the sheet, $N_{m}=1324$ hydrogen atoms can be attached (for every two internal carbon atoms there is one hydrogen atom). Thus, the graphene sheet under consideration can be described by the formula $C_{N_{c}}H_{N_{b}}=C_{2798}H_{148}$, and the corresponding graphone sheet by the formula $C_{N_{c}}$$H_{N_{b}+N_{m}}=C_{2798}H_{1472}$. If $0\leq N_{h}\leq N_{m}$ of hydrogen atoms are one-side attached to the graphene carbon atoms, the dimensionless concentration of attached hydrogen atoms (hydrogenation density) is $p=N_{h}/\left(N_{c}-N_{b}\right)\in \left[0,0.5\right]$.

The size of the graphene sheet was chosen on the basis of the possibilities of computer modeling. For the larger sheet, we can reach higher precision in determining the maximal possible density of the one-side hydrogenation (fluorination or chlorination). On the other hand, the increase in the sheet size results in considerable increase in computation time. The used sheet size, 8.47×8.37 nm^2^, provides a compromise between the computation time and the resulting precision. The use of a sheet with half the area, with the size 6.02×5.81 nm^2^ which includes $N_{c}=1398$ carbon atoms, gives practically the same results.

For the modeling of random hydrogenation (fluorination, chlorination) of a graphene sheet, first we consider the perfect graphone sheet, namely the graphene sheet with $N_{m}$ hydrogen (fluorine, chlorine) atoms attached to its outer surface. Then we randomly remove $N_{0}$ atoms and get a sheet with dimensionless hydrogenation density $p=\left(N_{m}-N_{0}\right)/\left(N_{c}-N_{b}\right)$. After that, having solved the problem for the minimum of the energy (Equation 8), we find possible stationary structures of the modified sheet. Each structure (atomic packaging) will be characterized by the specific energy $E_{c}=E/N_{c}$. To evaluate this energy, the removal of $N_{0}$ atoms will be carried out by 128 independent random ways. This allows you to find the average value and standard deviation of the specific energy over 128 independent random implementations of the hydrogenation of the sheet with a fixed attachment density *p*.

A free graphone sheet can have two main structures: the single roll and the double roll [11,22]. Two more stable structures are possible on a flat substrate: a flat form of a sheet placed parallel to the surface of the substrate and a convex form of the sheet partially torn off the substrate with folded edges attached to the substrate. The characteristic appearance of these four stable structures of a sheet on flat substrate is shown in Figure 1. For the flat form, the sheet is located parallel to the substrate plane and hydrogen atoms are randomly attached to its outer side. The structure of a single roll has the form of densely-packed roll (scroll) of a sheet with an external hydrogenated side, lying on a flat substrate. Double roll structure is realized by folding of the sheet into two scrolls simultaneously from two opposite edges.

Dependencies of the normalized energy of the sheet $E_{c}$ on the density of its hydrogenation *p* for four main structures of a sheet located on the flat surface of a graphite crystal and on (111) surface of nickel crystal are shown in Figure 2. On the surface of graphite, pure graphene sheet (hydrogenation density p=0) has only one stable flat structure. Sustainable roll structure can exist only for the hydrogenation density p>0.018, a stable double roll structure can exist only for p>0.75, and partially convex structure can exist only for p>0.26. For lower hydrogenation density (for p<0.018, 0.075, 0.26), these structures become unstable on the surface of graphite and the graphene takes the form of a flat sheet.

The flat sheet form remains stable for p∈[0,0.49]. With full hydrogenation (for p=0.5), the flat form becomes unstable. The flat structure is most energetically favorable only for hydrogenation densities p∈[0,0.21], single roll structure—for p∈[0.21,0.41], and the double roll structure—for p∈[0.41,0.5]. Therefore, when a graphene sheet is located on a flat surface of crystalline graphite, it is hardly possible to achieve its one-sided hydrogenation with density p>0.21 because of the folding of the sheet into a roll.

The surface of crystalline nickel has a higher energy of the interaction with graphene sheet. Therefore, the flat form of the graphene sheet remains stable in this case for any density of hydrogenation of its outer side (for p∈[0,0.5]). The flat structure is energetically favorable for p<0.47. Roll folding of the sheet will be stable only for hydrogenation density p>0.038, and for p∈[0.47,0.5] it becomes the most energetically favorable, see Figure 2b. Double roll form can exist for any hydrogenation density p∈[0,0.5], but it will always be energetically disadvantageous. The partially convex sheet structure is always the most energetically disadvantageous, and it is stable for hydrogenation density p>0.32. Therefore, the location of the graphene sheet on a flat surface of nickel crystal substrate allows to achieve its hydrogenation with the density close to maximal possible.

Four similar stable structures of the graphene sheet are obtained by its unilateral fluorination (attaching fluorine atoms to the outer surface of the sheet with density p∈[0,0.5]). The characteristic appearance of these four stable structures of fluorinated graphene sheet on flat substrate is shown in Figure 3. Dependencies of the normalized energy of the sheet $E_{c}$ on the density of its fluorination *p* for four main structures of the sheet located on the flat surface of graphite crystal and on (111) surface of the nickel crystal are shown in Figure 4. Stable roll structures can exist on these substrates only for fluorination density p>0.018.

On the surface of crystalline graphite, the planar structure retains its stability for fluorination density p∈[0,0.42]. With higher fluorination density, the flat form becomes unstable. The partially convex shape of the sheet is stable only for p∈[0.3,0.5]. The flat structure is the most energetically favorable only for fluorination density p∈[0,0.20], the single roll structure – for density 0.20<p<0.35, and the double roll structure – for density 0.35<p≤0.5, see Figure 4a. Therefore, when a graphene sheet is located on a flat surface of crystalline graphite, it is impossible to achieve its one-side fluorination with density p>0.20 (this will be prevented by the folding of the sheet into a roll).

On the surface of crystalline nickel, the flat structure of the sheet remains stable for fluorination density p∈[0,0.5], and partially convex form remains stable only for 0.30<p≤0.5. Here, the flat shape is the most energetically favorable for fluorination density p∈[0,0.30], and the double roll structure—for 0.30<p≤0.5, see Figure 4b. Therefore by placing graphene sheet on a flat surface of a nickel crystal, the higher density of one-side fluorination, p<0.30, can be achieved. The decrease of the maximal fluorination density with respect to the maximally achievable hydrogenation density is related with the larger size of the fluoride atom.

Chlorine atoms are larger than the fluorine and hydrogen atoms, which should hinder the chlorination of graphene. Characteristic view of four stable structures of the chlorinated graphene sheet on a flat substrate shown in Figure 5. Dependencies of the normalized sheet energy $E_{c}$ on the chlorination density *p* for four main structures of a sheet located on the flat surface of a graphite crystal and on (111) the surface of the crystal nickel are shown in Figure 6. On these substrates, a stable single-roll structures can exist only for chlorination density p≥0.019.

On the surface of crystalline graphite, the planar structure retains its stability for p∈[0,0.25]. With higher chlorination densities, the flat form becomes unstable. The double-roll structure is stable only for p≥0.076, and the partially convex shape of the sheet is stable for p>0.13. The flat structure is the most energy-efficient only for chlorination densities p∈[0,0.16], a single-roll structure—for 0.16<p<0.285, and the double-roll structure—for p>0.285, see Figure 6a. Therefore, when a graphene sheet is located on a flat surface of crystalline graphite, only the one-side chlorination with the density p<0.16 can be achieved.

On the surface of crystalline nickel, the flat structure of the sheet remains stable for p∈[0,0.264]. For the higher chlorination, the flat form becomes unstable. The double-roll structure is stable only for p≥0.094, and the partially convex shape of the sheet is stable for p>0.15. The flat structure is the most energy-efficient for chlorination densities p∈[0,0.21], the single-roll structure—for 0.21<p<0.264, and the double-roll structure—for p>0.264, see Figure 6b. Therefore for the graphene sheet placed on a flat surface of nickel crystal, the one-side chlorination can only be achieved for densities p<0.21.

## 4. Conclusions

We have performed numerical modeling, based on the force field method, of mechanical conformations of graphene sheets placed on flat substrates, caused by the change in the density of one-side attachment of hydrogen, fluorine or chlorine atoms to the sheet. It is shown that the chemically modified graphene sheet can take four main forms on the flat substrate: the form of a flat sheet parallel to the surface of the substrate, the form of a sheet with convex shape partially detached from the substrate with bent edges attached to the substrate, and the form of single or double rolls on the substrate. On the surface of crystalline graphite, the flat sheet form is the most favorable in energy for hydrogenation density p<0.21, fluorination density p<0.20 and chlorination density p<0.16. The surface of crystalline nickel has higher energy of graphene adsorption, here the flat form of chemically modified sheet is the lowest in energy for hydrogenation density p<0.47, fluorination density p<0.30 and chlorination density p<0.21. Our modeling and quantitative estimates of the attachment densities can serve for the determination of the maximal possible one-side chemical modifications of the flat graphene sheets. At higher attachment densities, the rolling of the sheet will prevent it from further chemical modification.

## Figures and Tables

**Figure 1 materials-12-04179-f001:**
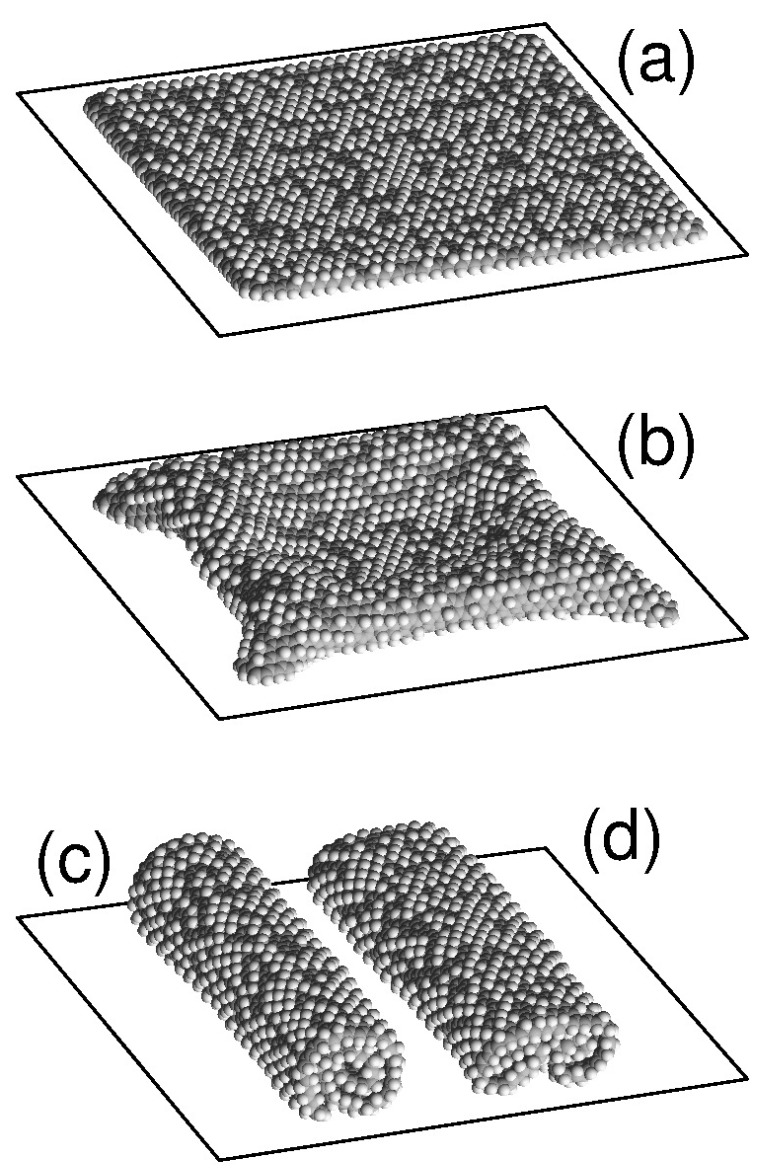
View of a square graphene sheet of size 8.47×8.37 nm^2^ (the number of carbon atoms $N_{c}=$2798), placed on a flat surface of a graphite crystal with attachment density of hydrogen atoms to graphene outer surface p=0.3019 (the number of hydrogen atoms attached to the surface is $N_{h}=800$) in (**a**) planar structure parallel to the substrate, (**b**) convex structure with edges attached to the substrate, and in structures of (**c**) single roll and (**d**) double roll. Dark (light) beads show carbon (hydrogen) atoms.

**Figure 2 materials-12-04179-f002:**
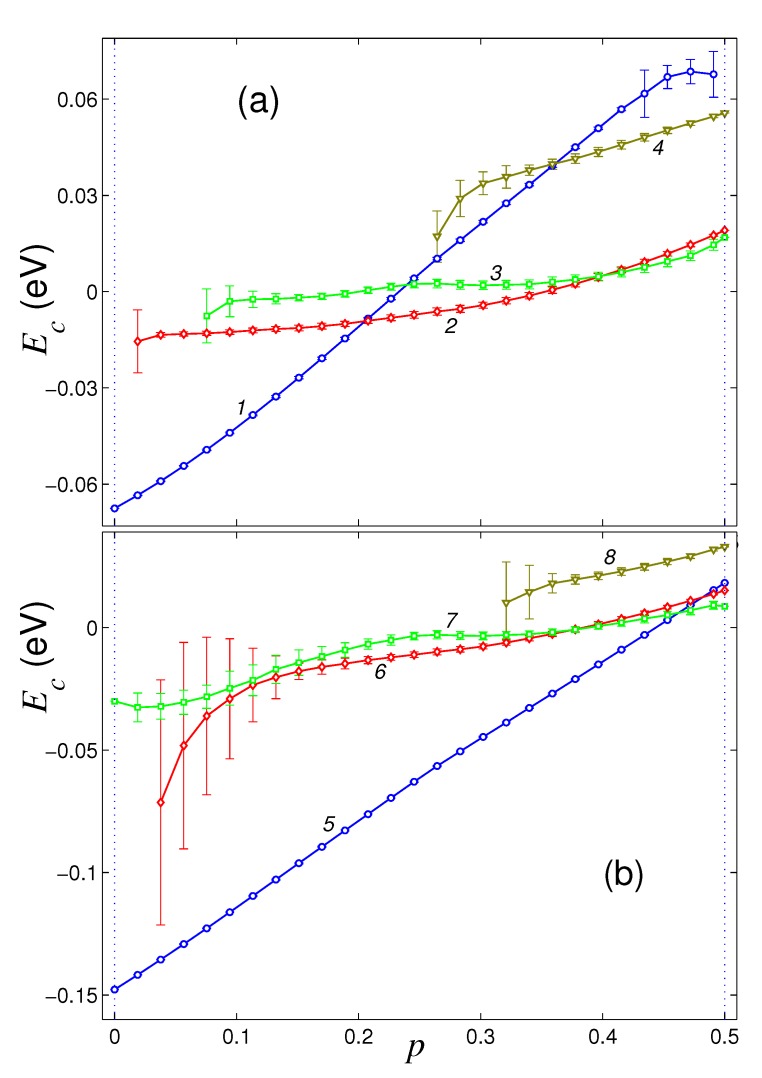
The dependence of the normalized energy $E_{c}=E/N_{c}$ of a square graphene sheet of size 8.47×8.37 nm^2^ on the dimensionless hydrogenation density *p* for a sheet located on a flat surface of (**a**) graphite crystal and (**b**) (111) surface of nickel crystal. Curves 1, 5 give the dependencies for the flat structure, curves 2, 6 and 3, 7—for the single and double roll structures, curves 4, 8—for partially convex structure with the edges attached to the substrate.

**Figure 3 materials-12-04179-f003:**
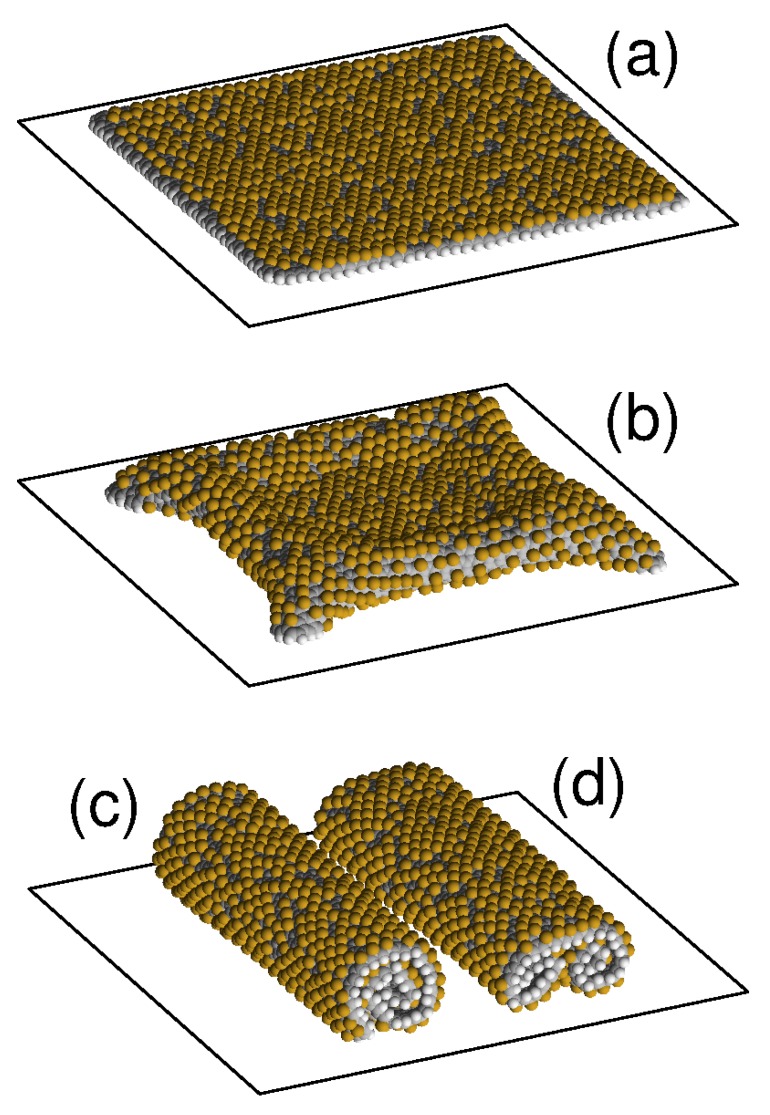
View of a square graphene sheet located on flat surface of nickel crystal with density of fluorine atoms attached to its outer side p=0.3396 (the number of fluorine atoms is $N_{h}=900$) in (**a**) planar structure parallel to the substrate, (**b**) convex structure with the edges attached to the substrate, and in structures of (**c**) single roll and (**d**) double roll. Gray/white/yellow beads show carbon/hydrogen/fluorine atoms.

**Figure 4 materials-12-04179-f004:**
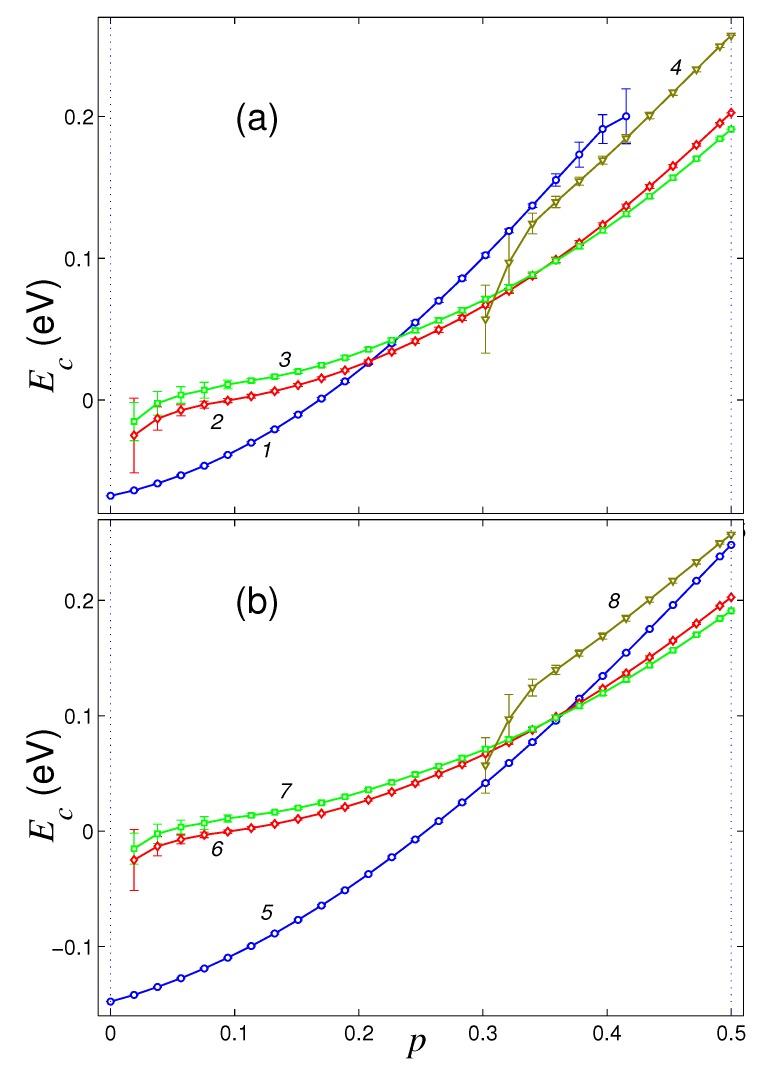
Dependence of the normalized energy $E_{c}$ of a square graphene sheet on the dimensionless fluorination density *p* for a sheet located on (**a**) flat surface of graphite crystal and (**b**) (111) surface of nickel crystal. Curves 1, 5 give the dependencies for the flat structure, curves 2, 6 and 3, 7—for the single and double roll structures, curves 4, 8—for partially convex structure with the edges attached to the substrate.

**Figure 5 materials-12-04179-f005:**
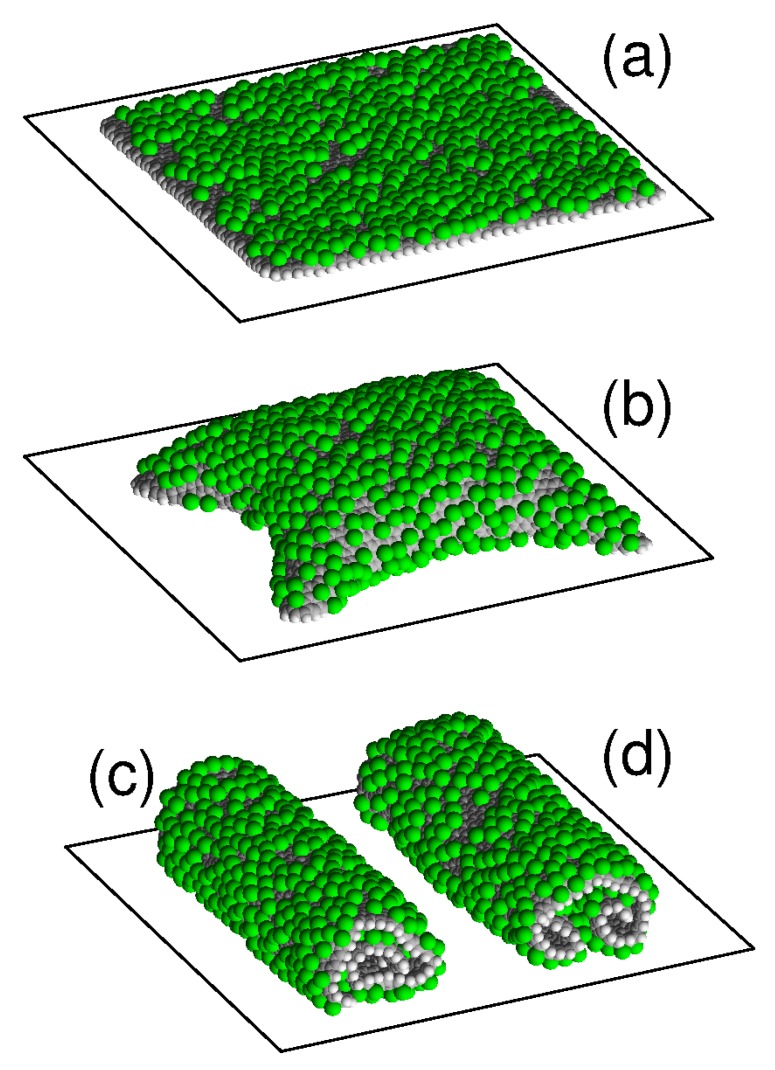
View of square graphene sheet located on a flat surface of nickel crystal at attachment density of chlorine atoms to its outer side p=0.2075 (the number of chlorine atoms is $N_{h}=550$) in (**a**) planar structure parallel to the substrate, (**b**) convex structure with the edges adjacent to the substrate, and in structures of (**c**) single roll and (**d**) double roll. Gray/white/green beads show carbon/hydrogen/chlorine atoms.

**Figure 6 materials-12-04179-f006:**
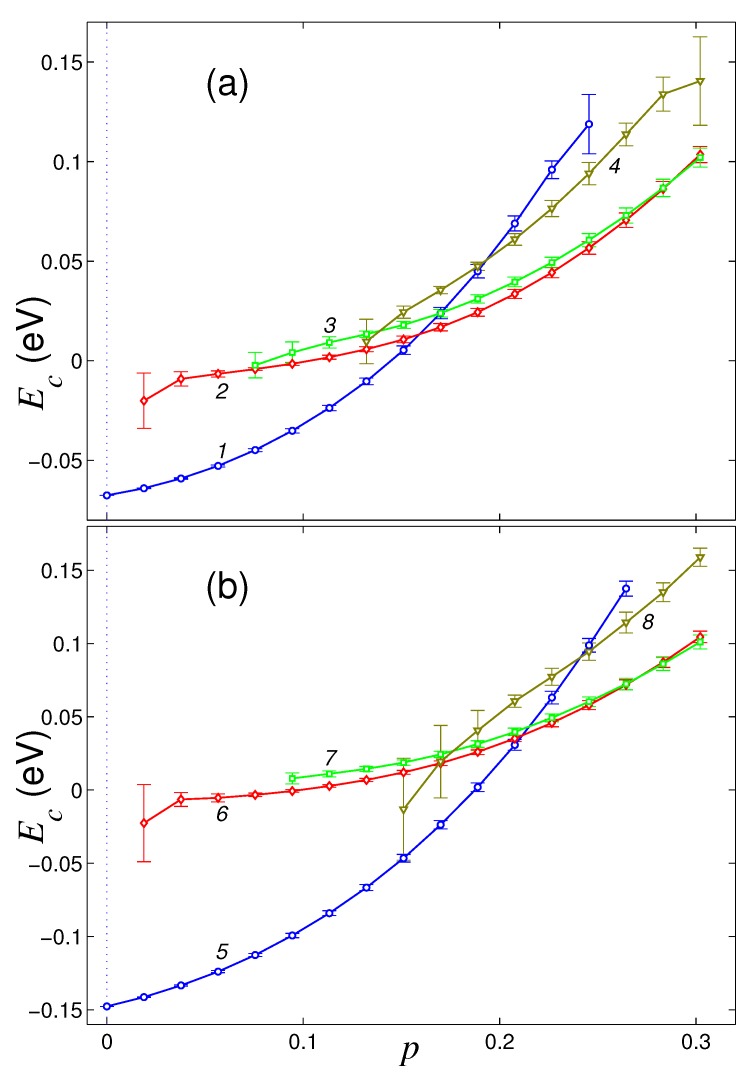
Dependence of the normalized energy $E_{c}$ of a square graphene sheet on the dimensionless chlorination density *p* for a sheet located on (**a**) flat surface of graphite crystal and (**b**) (111) surface of nickel crystal. Curves 1, 5 give the dependencies for the flat structure, curves 2, 6 and 3, 7—for the single and double roll structures, curves 4, 8—for partially convex structure with the edges attached to the substrate.

**Table 1 materials-12-04179-t001:** Values of the Morse potential parameters (Equation 1) for different valence bonds X—Y (C and C^′^ are carbon atoms involved in the formation of the sp^2^ and sp^3^ bonds).

X—Y	$\epsilon_{b}$ (eV)	$\rho_{0}$ (Å)	α (Å^−1^)
C—C	4.9632	1.418	1.7889
C—C^′^	4.0	1.522	1.65
C—H	4.28	1.08	1.8
C^′^—F	5.38	1.36	2.0
C^′^—Cl	3.40	1.761	2.0

**Table 2 materials-12-04179-t002:** Values of the parameters of the potential of the valence angle X–Y–Z (Equation 2) for different atoms (atom W = H, F, Cl).

X—Y—Z	$\epsilon_{a}$ (eV)	$\varphi_{0}$ (°)
C—C—C	1.3143	120.0
C—C^′^—C	1.3	109.5
C^′^—C^′^—C^′^	1.3	109.5
C—C—H	0.8	120.0
C—C^′^—W	1.0	109.5
C^′^—C^′^—H	1.0	109.5
H—C^′^—H	0.7	109.5

**Table 3 materials-12-04179-t003:** Values of the parameters of the Lennard-Jones potential (Equation 5) for different pairs of interacting atoms X, Y.

X,Y	$\epsilon_{0}$ (eV)	σ (Å^−1^)
C,C	0.002757	3.393
H,H	0.000681	2.471
C,H	0.001369	2.932
F,F	0.002645	3.118
C,F	0.002700	3.256
Cl,Cl	0.009843	3.516
C,Cl	0.005209	3.455

**Table 4 materials-12-04179-t004:** Values of the parameters of the substrate potential $W_{s}\left(h\right)$ for different flat substrates (harmonic coupling parameter $K_{0}=W_{s}^{\prime \prime }\left(h_{0}\right)$, frequency gap $\omega_{0}=\sqrt{K_{0}/2M}$ ).

	$\epsilon_{s}$ (eV)	$h_{0}$ (Å^−1^)	*l*	*k*	$K_{0}$ (N/m)	$\omega_{0}$ (cm^−1^)
SiO_2_	0.037	4.13	16	3.75	2.1	54.5
graphite	0.052	3.37	10	3.75	2.8	62.4
Si (100)	0.061	3.85	14	3.75	3.4	69.8
6H-SiC	0.073	4.19	17	3.75	4.2	77.5
Au (111)	0.073	2.96	10	3.5	4.7	81.4
h-BN	0.090	3.46	10	3.75	4.5	80.1
Ni (111)	0.133	2.14	-	-	41.0	240

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
