# Peer review of "Modeling of One-Side Surface Modifications of Graphene"

_materials, 2019, doi:10.3390/ma12244179_

Round 1

Reviewer 1 Report

In the manuscript entitled “One-side surface modifications of graphene” the author present numerical simulations of the mechanical properties of graphene nano-sheets when placed on top of various substrates. More precisely, they investigate the stability of the various structures the sheet can form in the presence of one-side chemical modification by hydrogen, fluorine and chlorine. Furthermore, they investigate the chemical modifications due the presence of much sizable entities such CH3 and C6H5 molecules.  

I find the manuscript well-written in most parts, but I think there are a number of issues that the authors should fix before I can make my final decision.

The authors should commend on the chosen size of the system. Why a total number of Nc=2798 carbon atoms was used? What’s the justification for such selection? And how a different size system would affect the results of the simulation? If possible, a comparison between different system’s sizes should be made and discussed in the manuscript?

Through the manuscript the authors mention some limitations on the density p for the various structures, for example p>0.018 for the “rolled structure” in line 142, but it is not clear to me why such limitations exist. What happens to a particular structure when the density exceeds the stated critical value?

The manuscript up to section 3 is well-written and the results are adequately presented and discussed. However, I find the sections 4 and 5 rather weak, very brief and thus not convincing. It’s not clearly explained how the results were obtained and how the conclusions about the structures were drawn. The authors should add figures -similar to Figs. 2,4,6- to support the discussion and the conclusions on the sections 3 and 4.

Author Response

Response to Reviewer 1 Comments

Point 1: The authors should commend on the chosen size of the system. Why a total number of Nc=2798 carbon atoms was used? What’s the justification for such selection? And how a different size system would affect the results of the simulation? If possible, a comparison between different system’s sizes should be made and discussed in the manuscript?

Response 1: The rationale is added in the text.

The size of the graphene sheet was chosen on the basis of the possibilities of computer modeling. For the larger sheet, we can reach higher precision in determining the maximal possible density of the one-side hydrogenation (fluorination or  chlorination). On the other hand, the increase in the sheet size results in considerable increase in computation time. The used sheet size,  8.47x8.37 nm2, provides a compromise between the computation time and the resulting precision. The use of a sheet with half the area, with the size  6.02x5.81 nm2 which includes Nc=1398 carbon atoms, gives practically the same results.

Point 2: Through the manuscript the authors mention some limitations on the density p for the various structures, for example p>0.018 for the “rolled structure” in line 142, but it is not clear to me why such limitations exist. What happens to a particular structure when the density exceeds the stated critical value?

Response 2: The explanation is added in the text.

For lower hydrogenation density (for p<0.018), these structures become unstable on the surface of graphite and the graphene takes the form of a flat sheet. 

Point 3: The manuscript up to section 3 is well-written and the results are adequately presented and discussed. However, I find the sections 4 and 5 rather weak, very brief and thus not convincing. It’s not clearly explained how the results were obtained and how the conclusions about the structures were drawn. The authors should add figures -similar to Figs. 2,4,6- to support the discussion and the conclusions on the sections 3 and 4.

Response 3: We agree with the Referee that the sections 4 and 5 are not finalized, and we have deleted these sections in the amended version of the paper. 

Reviewer 2 Report

The paper is overall well written and the methodology used is overall appropriate. Issues I think would improve the paper include:

(1) The work is based on modelling, but this is not evident from the title or abstract. Please change abstract (ideally also title) to indicate this clearly.

(2) Methodology uses force-fields which are known to have issues in some cases. Whilst I have no reason to conclude that the force-field used generates wrong results, I would prefer to see a short paragraph which states this limitation. This is particularly important since the force-field used in this case is what one which has not yet been widely used. Maybe it would be appropriate to add a comparison to one of the more widely used force-fields, such as AIREBO.

(3) The paper would be clearer is some of the figures were placed next to each other to aid comparison. Colours may be not easy to interpret, particularly  for individuals who have difficulty in distinguishing between green and red.

(4) Add a paragraph on strengths and limitations of this work. This would greatly improve the quality of the work.

(5) Please consider looking at https://onlinelibrary.wiley.com/doi/full/10.1002/andp.201700330 for another example of conformations of graphene and properties.

(6) Conclusion should be improved to make it more clear what main findings where. 

Author Response

Response to Reviewer 2 Comments

Point 1: The work is based on modelling, but this is not evident from the title or abstract.

Please change abstract (ideally also title) to indicate this clearly.

Response 1: We have changed the title of the paper and edited the Abstract and Introduction.

Point 2: Methodology uses force-fields which are known to have issues in some cases. Whilst I have no reason to conclude that the force-field used generates wrong results, I would prefer to see a short paragraph which states this limitation. This is particularly important since the force-field used in this case is what one which has not yet been widely used.

Maybe it would be appropriate to add a comparison to one of the more widely used force-fields, such as AIREBO.

Response 2: In our modeling, the AMBER (J. Am. Chem. Soc. V.117(19), 1995) force field was used. The results were also checked with the use of the force field COMPASS. Since in all the available force fields the embedded interaction parameters are carefully checked, we expect that all the force fields, including the AIREBO, will give very close results. We give a reference to the force field AMBER.

Point 3: The paper would be clearer is some of the figures were placed next to each other to aid comparison. Colours may be not easy to interpret, particularly  for individuals who have difficulty in distinguishing between green and red.

Response 3: The quality of line indexing in Figs. 2, 4 and 6 is improved.

Point 4: Add a paragraph on strengths and limitations of this work. This would greatly improve the quality of the work.

Response 4: We rewrote the Сonclusion.

Point 5: Please consider looking at https://onlinelibrary.wiley.com/doi/full/10.1002/andp.201700330 for another example of conformations of graphene and properties.

Response 5: The reference to  [32] (\bibitem{grima18}) is added in the amended text.

Point 6: Conclusion should be improved to make it more clear what main findings where.

Response 6: The following sentence is added in Conclusions:

Our modeling and quantitative estimates of the attachment densities can serve for the determination of the maximal possible one-side chemical modifications of the flat graphene sheets, when at higher attachment densities the rolling of the sheet will prevent it from further chemical modification.

Reviewer 3 Report

In this paper, the authors perform numerical modeling of the mechanical conformations of graphene sheets placed on flat substrates, resulting from the change in the density of one-side attachment of hydrogen, fluorine or chlorine atoms to the sheet. They also consider mechanical conformations of the graphene sheets with one-side attachment of molecular groups like CH3, CH2–CH3, C6H5 (benzene ring) or OH (hydroxyl group).

The study is interesting from a fundamental viewpoint. The study seems well-conducted and is clearly communicated. The simulation results, although difficult to control, seem plausible.

The paper could be suitable for publication in the journal after a revision.

The authors should better motivate their research. The study is interesting from a theoretical viewpoint, but which is its practical importance? Also, I advise that the authors search and mention if their conclusions have any experimental validation.

To study the effect of one-side attachment of molecular groups the authors consider two types of couplings with the substrate. They mention that the strongest coupling is for graphene on Ni, while the weaker coupling applies to crystalline silicon, silicon carbide 6H-SiC, silver or gold. However, many useful optoelectronic or sensor applications of graphene are achieved with graphene on an insulating substrate such as SiO2, h-BN, HfO2, etc. Could the authors include such substrates?

Author Response

Response to Reviewer 3 Comments

Point 1: The authors should better motivate their research. The study is interesting from a theoretical viewpoint, but which is its practical importance? Also, I advise that the authors search and mention if their conclusions have any experimental validation.

Response 1: The following sentence is added in Conclusions:

Our modeling and quantitative estimates of the attachment densities can serve for the determination of the maximal possible one-side chemical modifications of the flat graphene sheets, when at higher attachment densities the rolling of the sheet will prevent it from further chemical modification.

Point 2: To study the effect of one-side attachment of molecular groups the authors consider two types of  couplings with the substrate. They mention that the strongest coupling is for graphene on Ni, while the weaker coupling applies to crystalline silicon, silicon carbide 6H-SiC, silver or gold. However, many useful optoelectronic or sensor applications of graphene are achieved with graphene  on an insulating substrate such as SiO2, h-BN, HfO2, etc. Could the authors include such substrates?

Response 2: The Table 4 is added with the values of the interaction parameters for different substrates, including the abovementioned.

Round 2

Reviewer 1 Report

The revised manuscript is clearly improved as compared to the first version. The authors provide adequate explanations/answers to my question. I believe the revised version of the manuscript should be accepted for publication in the Materials Journal. I can only suggest some minors edits into the manuscript to fix some typos etc. 

line 1: "We model, with the of the force field method,..." (similarly in line 204)

line 53: The acronym AMBER needs to be written out. 

lines 214-217: The statement "Our modeling... chemical modification" is unclear. Maybe it should be split into two full sentences. 

Author Response

line 1: "We model, with the of the force field method,..." (similarly in line 204)

fixed

line 53: The acronym AMBER needs to be written out. 

fixed

lines 214-217: The statement "Our modeling... chemical modification" is unclear. Maybe it should be split into two full sentences. 

fixed